# VideoMAE: Masked Autoencoders are Data-Efficient Learners for Self-Supervised Video Pre-Training

**Zhan Tong** [1,2*]   **Yibing Song** [2]   **Jue Wang** [2]   **Limin Wang** [1,3†]

[1]State Key Laboratory for Novel Software Technology, Nanjing University
[2]Tencent AI Lab       [3]Shanghai AI Lab

tongzhan@smail.nju.edu.cn   {yibingsong.cv, arphid}@gmail.com   lmwang@nju.edu.cn

## Abstract

Pre-training video transformers on extra large-scale datasets is generally required to achieve premier performance on relatively small datasets. In this paper, we show that video masked autoencoders (VideoMAE) are data-efficient learners for self-supervised video pre-training (SSVP). We are inspired by the recent Image-MAE [30] and propose customized video tube masking with an extremely high ratio. This simple design makes video reconstruction a more challenging and meaningful self-supervision task, thus encouraging extracting more effective video representations during the pre-training process. We obtain three important findings with VideoMAE: (1) An extremely high proportion of masking ratio (i.e., $90\%$ to $95\%$) still yields favorable performance for VideoMAE. The temporally redundant video content enables higher masking ratio than that of images. (2) VideoMAE achieves impressive results on very small datasets (i.e., around 3k-4k videos) without using any extra data. This is partially ascribed to the challenging task of video reconstruction to enforce high-level structure learning. (3) VideoMAE shows that data quality is more important than data quantity for SSVP. Domain shift between pre-training and target datasets is an important factor. Notably, our VideoMAE with the vanilla ViT backbone can achieve 87.4% on Kinects-400, 75.4% on Something-Something V2, 91.3% on UCF101, and 62.6% on HMDB51, without using any extra data. Code is available at https://github.com/MCG-NJU/VideoMAE.

## 1 Introduction

Transformer [70] has brought significant progress in natural language processing [17, 7, 54]. The vision transformer [20] also improves a series of computer vision tasks including image classification [66, 88], object detection [8, 37], semantic segmentation [80], object tracking [13, 16], and video recognition [6, 3]. The multi-head self-attention upon linearly projected image/video tokens is capable of modeling global dependency among visual content either spatially or temporally. The inductive bias is effectively reduced via this flexible attention mechanism.

Training effective vision transformers (ViTs) typically necessitates large-scale supervised datasets. Initially, the pre-trained ViTs achieve favorable performance by using hundreds of millions of labeled images [20]. For video transformers [3, 6], they are usually derived from image-based transformers and heavily depend on the pre-trained models from large-scale image data (e.g., ImageNet [57]). Previous trials [3, 6] on training video transformers from scratch yield unsatisfied results (except for MViT [21] with a strong inductive bias). Therefore, the learned video transformers are naturally biased by image-based models, and it still remains a challenge that *how to effectively and efficiently train a vanilla vision transformer on the video dataset itself without using any pre-trained model*

---

[*]Work is done during internship at Tencent AI Lab.   [†]Corresponding author.

36th Conference on Neural Information Processing Systems (NeurIPS 2022).

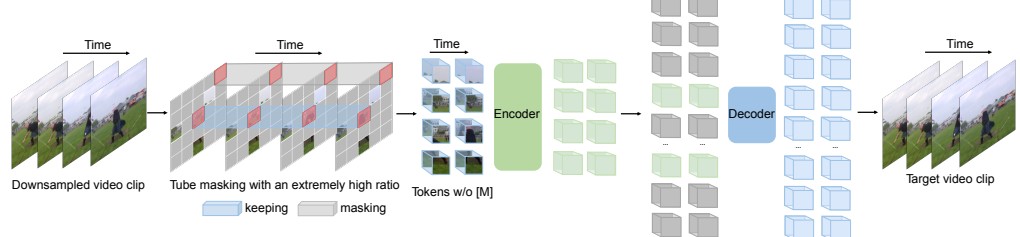

Figure 1: **VideoMAE** performs the task of masking random cubes and reconstructing the missing ones with an asymmetric encoder-decoder architecture. Due to high redundancy and temporal correlation in videos, we present the customized design of tube masking with an extremely high ratio (90% to 95%). This simple design enables us to create a more challenging and meaningful self-supervised task to make the learned representations capture more useful spatiotemporal structures.

*or extra image data.* Moreover, the existing video datasets are relatively small compared with image datasets, which further increases the difficulty of training video transformers from scratch. Meanwhile, self-supervised learning has shown remarkable performance by using large-scale image datasets [14, 9]. The learned representations have outperformed the ones via supervised learning when being transferred to downstream tasks. It is expected that this self-supervised learning paradigm can provide a promising solution to address the challenge of training video transformers.

Following the success of masked autoencoding in NLP [17] and images [30, 4], we present a new self-supervised video pre-training (SSVP) method, termed as *Video Masked Autoencoder* (VideoMAE). Our VideoMAE inherits the simple pipeline of masking random cubes and reconstructing the missing ones. However, the extra time dimension of videos makes them different from images in this masked modeling. First, video frames are often densely captured, and their semantics varies slowly in time [87]. This temporal redundancy would increase the risk of recovering missing pixels from the spatiotemporal neighborhood with little high-level understanding. Furthermore, video could be viewed as the temporal evolution of static appearance, and there exists a correspondence between frames. This temporal correlation could lead to information leakage (i.e., masked spatiotemporal content re-occurrence) during reconstruction unless a specific masking strategy is considered. In this sense, for each masked cube, it is easy to find a corresponding and unmasked copy in adjacent frames. This property would make the learned models identify some "shortcut" features that are hard to generalize to new scenarios.

To make video masked modeling more effective, in this paper, we present a customized design of tube masking with an extremely high ratio in our VideoMAE. First, due to temporal redundancy, we use an *extremely high* masking ratio to drop the cubes from the downsampled clips. This simple strategy not only effectively increases the pre-training performance but also greatly reduces the computational cost due to the asymmetric encoder-decoder architecture. Second, to consider temporal correlation, we devise a simple yet effective *tube masking* strategy, which turns out to be helpful in relieving the risk of information leakage for cubes with no or negligible motion during reconstruction. With this simple yet effective design in our VideoMAE, we are able to successfully train vanilla ViT backbones on the relatively small-scale video datasets such as Something-Something [25], UCF101 [60], and HMDB51 [34], which significantly outperform the previous state of the art under the setting without extra data. In summary, the main contribution of this paper is threefold:

- We present a simple but effective video masked autoencoder that unleashes the potential of vanilla vision transformer for video recognition. To the best of our knowledge, this is the first masked video pre-training framework of simply using plain ViT backbones. To relieve the information leakage issue in masked video modeling, we present the tube masking with an extremely high ratio, which brings the performance improvement to the VideoMAE.

- Aligned with the results in NLP and Images on masked modeling, our VideoMAE demonstrates that this simple masking and reconstruction strategy provides a good solution to self-supervised video pre-training. The models pre-trained with our VideoMAE significantly outperform those trained from scratch or pre-trained with contrastive learning methods.

- We obtain extra important findings on masked modeling that might be ignored in previous research in NLP and Images. (1) We demonstrate that VideoMAE is a data-efficient learner that could be successfully trained with only 3.5k videos. (2) Data quality is more important than quantity for SSVP when a domain shift exists between the source and target dataset.

## 2 Related Work

**Video representation learning.** Learning good video representations has been heavily investigated in the literature. The supervised learning methods [58, 75, 69, 10, 6] usually depend on the image backbones. The video encoder backbones are first pre-trained with image data in a supervised form. Then, these backbones are fine-tuned on the video dataset for classifying human actions. Meanwhile, some methods [67, 22, 21] directly train video backbones from videos in a supervised manner. Besides supervised learning, semi-supervised video representation learning has also been studied [59]. The representations of labeled training samples are utilized to generate supervision signals for unlabeled ones. Supervised or semi-supervised representation learning mainly uses a top-down training paradigm, which is not effective in exploring the inherent video data structure itself. Meanwhile, some multimodal contrastive learning methods [36, 42, 62] have been developed to learn video representation from noisy text supervision.

For self-supervised learning, the prior knowledge of temporal information has been widely exploited to design pretext tasks [78, 44, 82, 5] for SSVP. Recently, contrastive learning [28, 45, 29, 52, 24, 27] is popular to learn better visual representation. However these methods heavily rely on strong data augmentation and large batch size [23]. Predicting the video clip with autoencoders in pixel space has been explored for representation learning by using CNN or LSTM backbones [48, 61], or conducting video generation with autoregressive GPT [83]. Instead, our VideoMAE aims to use the simple masked autoencoder with recent ViT backbones to perform data-efficient SSVP.

**Masked visual modeling.** Masked visual modeling has been proposed to learn effective visual representations based on the simple pipeline of masking and reconstruction. These works mainly focus on the image domain. The early work [72] treated the masking as a noise type in denoised autoencoders [71] or inpainted missing regions with context [47] by using convolutions. iGPT [11] followed the success of GPT [7, 55] in NLP and operated a sequence of pixels for prediction. The original ViT [20] investigated the masked token prediction for self-supervised pre-training. More recently, the success of vision transformer has led to investigation of Transformer-based architectures for masked visual modeling [4, 19, 30, 79, 81, 89]. BEiT [4], BEVT [76] and VIMPAC [64] followed BERT [17] and proposed to learn visual representations from images and videos by predicting the discrete tokens [56]. MAE [30] introduced an asymmetric encoder-decoder architecture for masked image modeling. MaskFeat [79] proposed to reconstruct the HOG features of masked tokens to perform self-supervised pre-training in videos. VideoMAE is inspired by the ImageMAE and introduces specific design in implementation for SSVP. In particular, compared with previous masked video modeling [30, 76, 64], we present a simpler yet more effective video masked autoencoder by directly reconstructing the pixels. Our VideoMAE is the first masked video pre-training framework of simply using plain ViT backbones.

## 3 Proposed Method

In this section, we first revisit ImageMAE [30]. Then we analyze the characteristics of video data. Finally, we show how we explore MAE in the video data by presenting our VideoMAE.

### 3.1 Revisiting Image Masked Autoencoders

ImageMAE [30] performs the masking and reconstruction task with an asymmetric encoder-decoder architecture. The input image $I \in \mathcal{R}^{3 \times H \times W}$ is first divided into regular non-overlapping patches of size $16 \times 16$, and each patch is represented with *token embedding*. Then a subset of tokens are randomly masked with a high masking ratio (75%), and only the remaining ones are fed into the transformer *encoder* $\Phi_{\text{enc}}$. Finally, a shallow *decoder* $\Phi_{\text{dec}}$ is placed on top of the visible tokens from the encoder and learnable mask tokens to reconstruct the image. The *loss function* is mean squared error (MSE) loss between the normalized masked tokens and reconstructed ones in the pixel space:

$$\mathcal{L} = \frac{1}{\Omega} \sum_{p \in \Omega} |I(p) - \hat{I}(p)|^2, \tag{1}$$

where $p$ is the token index, $\Omega$ is the set of masked tokens, $I$ is the input image, and $\hat{I}$ is the reconstructed one.

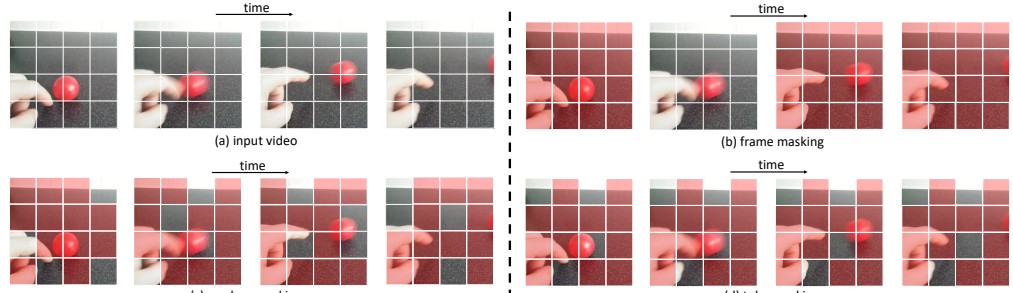

Figure 2: Slowness is a general prior in (a) video data [87]. This leads to two important characteristics in time: temporal redundancy and temporal correlation. Temporal redundancy makes it possible to recover pixels under an extremely high masking ratio. Temporal correlation leads to easily reconstruct the missing pixels by finding those corresponding patches in adjacent frames under plain (b) frame masking or (c) random masking. To avoid this simple task and encourage learning representative representation, we propose a (d) tube masking, where the masking map is the same for all frames.

## 3.2 Characteristics of Video Data

Compared with static images, video data contain temporal relations. We show the motivation of our VideoMAE by analyzing video characteristics.

**Temporal redundancy.** There are frequently captured frames in a video. The semantics vary slowly in the temporal dimension [87]. We observe that consecutive frames are highly redundant, as shown in Figure 2. This property leads to two critical issues in masked video autoencoding. First, it would be less efficient to keep the original temporal frame rate for pre-training. This would draw us to focus more on static or slow motions in our masked modeling. Second, temporal redundancy greatly dilutes motion representations. This would make the task of reconstructing missing pixels not difficult under the normal masking ratio (e.g., 50% to 75%). The encoder backbone is not effective in capturing motion representations.

**Temporal correlation.** Videos could be viewed as the temporal extension of static appearance, and therefore there exists an inherent correspondence between adjacent frames. This temporal correlation could increase the risk of information leakage in the masking and reconstruction pipeline. In this sense, as shown in Figure 2, we can reconstruct the masked patches by finding the spatiotemporal corresponding unmasked patches in the adjacent frames under plain random masking or frame masking. In this case, it might guide the VideoMAE to learn low-level temporal correspondence rather than high-level information such as spatiotemporal reasoning over the content. To alleviate this behavior, we need to propose a new masking strategy to make the reconstruction more challenging and encourage effective learning of spatiotemporal structure representations.

## 3.3 VideoMAE

To relieve the above issues in video masked modeling, we make the customized design in our VideoMAE, and the overall pipeline is shown in Figure 1. Our VideoMAE takes the *downsampled frames* as inputs and uses the *cube embedding* to obtain video tokens. Then, we propose a simple design of *tube masking with high ratio* to perform MAE pre-training with an asymmetric encoder-decoder architecture. Our backbone uses the vanilla ViT with *joint space-time attention*.

**Temporal downsampling.** According to the above analysis on temporal redundancy over consecutive frames, we propose to use the strided temporal sampling strategy to perform more efficient video pre-training. Formally, one video clip consisting of $t$ consecutive frames is first randomly sampled from the original video $V$. We then use temporal sampling to compress the clip to $T$ frames, each of which contains $H \times W \times 3$ pixels. In experiments, the stride $\tau$ is set to 4 and 2 on Kinetics and Something-Something, respectively.

**Cube embedding.** We adopt the joint space-time cube embedding [3, 21, 38] in our VideoMAE, where we treat each cube of size $2 \times 16 \times 16$ as one token embedding. Thus, the cube embedding layer obtains $\frac{T}{2} \times \frac{H}{16} \times \frac{W}{16}$ 3D tokens and maps each token to the channel dimension $D$. This design can decrease the spatial and temporal dimension of input, which helps to alleviate the spatiotemporal redundancy in videos.

**Tube masking with extremely high ratios.** First, temporal redundancy is a factor affecting Video-MAE design. We find that VideoMAE is in favor of extremely high masking ratios (e.g. 90% to 95%) compared with the ImageMAE. Video information density is much lower than images, and we expect a high ratio to increase the reconstruction difficulty. This high masking ratio is helpful to mitigate the information leakage during masked modeling and make masked video reconstruction a meaningful self-supervised pre-training task.

Second, temporal correlation is another factor in our VideoMAE design. We find even under the extremely high masking ratio, we can still improve the masking efficiency by proposing the temporal tube masking mechanism. Temporal tube masking enforces a mask to expand over the whole temporal axis, namely, different frames sharing the same masking map. Mathematically, the tube mask mechanism can be expressed as $\mathbb{I}[p_{x,y,\cdot} \in \Omega] \sim \text{Bernoulli}(\rho_{\text{mask}})$ and different time $t$ shares the same value. With this mechanism, temporal neighbors of masked cubes are always masked. So for some cubes with no or small motion (e.g., finger cube in 4th row of Figure 2 (d)), we can not find the spatiotemporal corresponding content in all frames. In this way, it would encourage our VideoMAE to reason over high-level semantics to recover these totally missing cubes. This simple strategy can alleviate the information leakage for cubes with no or negligible motion, and turns out to be effective in practice for masked video pre-training.

**Backbone: joint space-time attention.** Due to the high proportion of masking ratio mentioned above, only a few tokens are left as the input for the encoder. To better capture high-level spatio-temporal information in the remaining tokens, we use the vanilla ViT backbone [20] and adopt the joint space-time attention [3, 38]. Thus, all pair tokens could interact with each other in the multi-head self-attention layer [70]. The specific architecture design for the encoder and decoder is shown in supplementary materials. The quadratic complexity of the joint space-time attention mechanism is a computational bottleneck, while our design of an extremely high masking ratio alleviates this issue by only putting the unmasked tokens (e.g., 10%) into the encoder during the pre-training phase.

## 4 Experiments

### 4.1 Datasets

We evaluate our VideoMAE on five common video datasets: Kinetics-400 [33], Something-Something V2 [25], UCF101 [60], HMDB51 [34], and AVA [26]. The Kinetics-400 contains around 240k training videos and 20k validation videos of 10s from 400 classes. The Something-Something V2 is another large-scale video dataset, having around 169k videos for training and 20k videos for validation. In contrast to Kinetics-400, this dataset contains 174 motion-centric action classes. These two large-scale video datasets focus on different visual cues for action recognition. UCF101 and HMDB51 are two relatively small video datasets, which contain around 9.5k/3.5k train/val videos and 3.5k/1.5k train/val videos, respectively. Compared with those large-scale video datasets, these two small datasets are more suitable for verifying the effectiveness of VideoMAE, as training large ViT models is more challenging on small datasets. Moreover, we also transfer the learned ViT models by VideoMAE to downstream action detection task. We work on AVA, a dataset for spatiotemporal localization of human actions with 211k training and 57k validation video segments. In experiments of downstream tasks, we fine-tune the pre-trained VideoMAE models on the training set and report the results on the validation set. The implementation details are described in Appendix § B.

### 4.2 Ablation Studies

In this subsection, we perform in-depth ablation studies on VideoMAE design with the default backbone of 16-frame ViT-B on Something-Something V2 (SSV2) and Kinetics-400 (K400). The specific architectures for the encoder and decoder are shown in Appendix § A. For fine-tuning, we perform TSN [75] uniform sampling on SSV2 and dense sampling [77, 22] on K400. All models share the same inference protocol, i.e., 2 clips × 3 crops on SSV2 and 5 clips × 3 crops on K400.

**Decoder design.** The lightweight decoder is one key component of our VideoMAE. We conduct experiments with the different depths in Table 1a. Unlike in ImageMAE, a deep decoder here is important for better performance, while a shallow decoder could reduce the GPU memory consumption. We take 4 blocks for the decoder by default. The decoder width is set to half channel of the encoder (e.g., 384-d for ViT-B), following the design in the image domain.

| blocks | SSV2 | K400 | GPU mem. |
|---|---|---|---|
| 1 | 68.5 | 79.0 | 7.9G |
| 2 | 69.2 | 79.2 | 10.2G |
| 4 | **69.6** | **80.0** | 14.7G |
| 8 | 69.3 | 79.7 | 23.7G |

| case | ratio | SSV2 | K400 |
|---|---|---|---|
| tube | 75 | 68.0 | 79.8 |
| tube | 90 | **69.6** | **80.0** |
| random | 90 | 68.3 | 79.5 |
| frame | 87.5$^*$ | 61.5 | 76.5 |

| input | target | SSV2 | K400 |
|---|---|---|---|
| $T \times \tau$ | *center* | 63.0 | 79.3 |
| $T \times \frac{\tau}{2}$ | $T \times \frac{\tau}{2}$ | 68.9 | 79.8 |
| $T \times \tau$ | $T \times \tau$ | **69.6** | 80.0 |
| $T \times \tau$ | $2T \times \frac{\tau}{2}$ | 69.2 | **80.1** |

(a) **Decoder depth**. 4 blocks of decoder achieve the best trade-off. "GPU mem." is GPU memory during pre-training, benchmarked in one GPU with a batch size of 16.

(b) **Mask sampling**. We compare different masking strategies. Our proposed tube masking with an extremely high ratio works the best. $*$"87.5" means masking 14/16 frames.

(c) **Reconstruction target**. $T \times \tau$ denotes "frames $\times$stride". *center* denotes the center frame of the input clip. $T$ is set to 16 as default. $\tau$ is set to 2 and 4 on SSV2 and K400, respectively.

| case | SSV2 | K400 |
|---|---|---|
| *from scratch* | 32.6 | 68.8 |
| ImageNet-21k sup. | 61.8 | 78.9 |
| IN-21k+K400 sup. | 65.2 | - |
| VideoMAE | **69.6** | **80.0** |

| dataset | method | SSV2 | K400 |
|---|---|---|---|
| IN-1K | ImageMAE | 64.8 | 78.7 |
| K400 | VideoMAE | 68.5 | **80.0** |
| SSV2 | VideoMAE | **69.6** | 79.6 |

| case | SSV2 | K400 |
|---|---|---|
| L1 loss | 69.1 | 79.7 |
| MSE loss | **69.6** | **80.0** |
| Smooth L1 loss | 68.9 | 79.6 |

(d) **Pre-training strategy**. Our VideoMAE works the best without using any extra data. "sup." is supervised training.

(e) **Pre-training dataset**. Our VideoMAE works the best when directly pre-training the models on the source datasets.

(f) **Loss function**. MSE loss works the best for the masking and reconstruction task in VideoMAE.

Table 1: Ablation experiments on **Something-Something V2** and **Kinetics-400**. Our backbone is 16-frame vanilla ViT-B and all models are pre-trained with *mask ratio* $\rho$=90% for 800 epochs, and fine-tuned for evaluation. We perform TSN [75] uniform sampling on SSV2 and dense sampling [77, 22] on K400. All models share the same inference protocol, i.e., 2 clips $\times$ 3 crops on SSV2 and 5 clips $\times$ 3 crops on K400. The default choice for our model is colored in gray .

**Masking strategy.** We compare different masking strategies in Table 1b. When increasing the masking ratio from 75% to 90% for tube masking, the performance on SSV2 boosts from 68.0% to 69.6%. Then, with an extremely high ratio, we find tube masking also achieves better performance than plain random masking and frame masking. We attribute these interesting observations to the redundancy and temporal correlation in videos. The conclusion on K400 is in accord with one on SSV2. One may note that the performance gap on K400 is lower than one on SSV2. We argue that the Kinetics videos are mostly stationary and scene-related. The effect of temporal modeling is not obvious. Overall, we argue that our default designs enforce the networks to capture more useful spatiotemporal structures and therefore make VideoMAE a more challenging task, which a good self-supervised learner hunger for.

**Reconstruction target.** First, if we only employ the center frame as the target, the results would decrease greatly as shown in Table 1c. The sampling stride is also sensitive. The result of small sampling stride $\frac{\tau}{2}$ is lower than default sampling stride $\tau$ (68.9% vs. 69.6% on SSV2). We also try to reconstruct $2T$ frames from the downsampled $T$ frames, but it obtains slightly worse results on SSV2. For simplicity, we use the input downsampled clip as our default reconstruction target.

**Pre-training strategy.** We compare different pre-training strategies in Table 1d. Similar to previous trials [3, 6], training video transformers from scratch yields unsatisfied results on video datasets. When pre-trained on the large-scale ImageNet-21K dataset, the video transformer obtains better accuracy from 32.6% to 61.8% on SSV2 and 68.8% to 78.9% on K400. Using the models pre-trained on both ImageNet-21K and Kinetics further increases accuracy to 65.2% on SSV2. Our VideoMAE can effectively train a video transformer on the video dataset itself without using any extra data and achieve the best performance (69.6% on SSV2 and 80.0% on K400).

**Pre-training dataset.** First, we pre-train the ViT-B on ImageNet-1K for 1600 epochs, following the recipes in [30]. Then we inflate the 2D patch embedding layer to our cube embedding layer following [10] and fine-tune the model on the target video datasets. The results surpass the model trained *from scratch* as shown in Table 1e. We also compare the ImageMAE pre-trained model with VideoMAE models pre-trained on video datasets. We see that our VideoMAE models can achieve better performance than ImageMAE. However, when we try to transfer the pre-trained VideoMAE models to the other video datasets (e.g. from Kinetics to Something-Something), the results are slightly worse than their counterpart, which is directly pre-trained on its own target video datasets. We argue that domain shift between pre-training and target datasets could be an important issue.

| dataset | training data | *from scratch* | MoCo v3 | VideoMAE |
|---------|---------------|----------------|---------|----------|
| K400 | 240k | 68.8 | 74.2 | **80.0** |
| Sth-Sth V2 | 169k | 32.6 | 54.2 | **69.6** |
| UCF101 | 9.5k | 51.4 | 81.7 | **91.3** |
| HMDB51 | 3.5k | 18.0 | 39.2 | **62.6** |

Table 2: Comparisons with the results of previous **self-supvised pre-training methods** on different datasets. We take 16-frame ViT-B as the default backbone. Notably, here MoCo v3 and VideoMAE all only use the *unlabelled* data in the training set of each dataset for pre-training and are all fine-tuned for evaluation.

| method | epoch | ft. acc. | lin. acc. | hours | speedup |
|--------|-------|----------|-----------|-------|---------|
| MoCo v3 | 300 | 54.2 | 33.7 | 61.7 | - |
| VideoMAE | 800 | **69.6** | 38.9 | 19.5 | **3.2×** |

Table 3: Comparisons with the **efficiency and effectiveness** on Something-Something V2. We report the fine-tuning (ft) and linear probing (lin) accuracy (%). The **wall-clock time** of pre-training is benchmarked in 64 Tesla V100 GPUs with PyTorch.

| method | K400 → SSV2 | K400 → UCF | K400 → HMDB |
|--------|-------------|------------|-------------|
| MoCo v3 | 62.4 | 93.2 | 67.9 |
| VideoMAE | **68.5** | **96.1** | **73.3** |

Table 4: Comparisons with the **feature transferability** on smaller datasets. We take 16-frame ViT-B as the default backbone. Notably, here MoCo v3 and VideoMAE are all pre-trained on Kinetics-400 with *unlabelled* data in the training set. Then the pre-trained model is fine-tuned on target datasets for evaluation.

**Loss function.** Table 1f contains an ablation study of loss function. We find that the MSE loss could achieve a higher result compared with the L1 loss and smooth L1 loss. Therefore, we employ the MSE loss by default.

## 4.3 Main Results and Analysis

**VideoMAE: data-efficient learner.** The self-supervised video pre-training (SSVP) has been extensively studied in previous works, but they mainly use the CNN-based backbones. Few works have investigated transformer-based backbone in SSVP. Therefore, to demonstrate the effectiveness of VideoMAE for transformer-based SSVP, we compare two methods implemented by ourselves: (1) training from scratch and (2) pre-training with contrastive learning (MoCo v3 [14]). For training from scratch, we carefully tune these hyper-parameters to successfully pre-train ViT-Base from the training set of the dataset. For pre-training with MoCo v3, we strictly follow the training practice in its image counterpart and carefully avoid the collapse issue.

The recognition accuracy is reported in Table 2. We see that our VideoMAE significantly outperforms other two training settings. For instance, on the largest dataset of Kinetics-400, our VideoMAE outperforms training from scratch by around 10% and MoCo v3 pre-training by around 5%. This superior performance demonstrates that masked autoencoder provides an effective pre-training mechanism for video transformers. We also see that the performance gap between our VideoMAE and the other two methods becomes larger as the training set becomes smaller. Notably, even with only 3.5k training clips on HMDB51, our VideoMAE pre-training can still obtain a satisfying accuracy (around 61%). This new result demonstrates that VideoMAE is a more data-efficient learner for SSVP. This property is particularly important for scenarios with limited data available and different with contrastive learning methods.

We compare the efficiency of VideoMAE pre-training and MoCo v3 pre-training in Table 3. The task of masked autoencoding with a high ratio is more challenging and thereby requires more training epochs (800 vs. 300). Thanks to the asymmetric encoder-decoder in our VideoMAE and extremely high masking ratio, our pre-training time is much shorter than MoCo v3 (19.5 vs. 61.7 hours).

**High masking ratio.** In VideoMAE, one core design is the extremely high masking ratio. We perform an investigation of this design on the Kinetics-400 and Something-Something V2 datasets. The results are shown in Figure 3. We see that the best masking ratio is extremely high, and even 95% can achieve good performance for both datasets. This result is difference from BERT [17] in NLP and MAE [30] in images. We analyze the temporal redundancy and correlation in videos makes it possible for our VideoMAE to learn plausible outputs with such a high masking ratio.

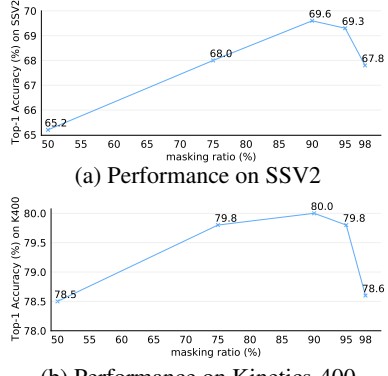

(a) Performance on SSV2

(b) Performance on Kinetics-400

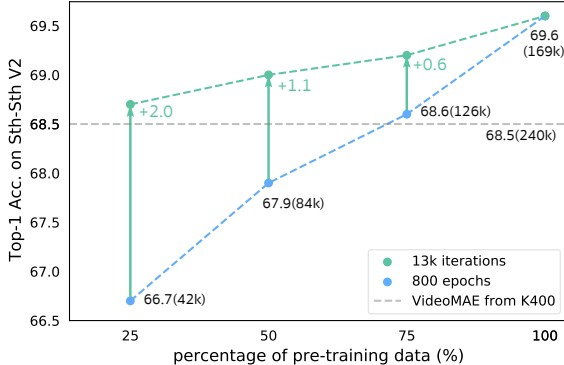

Figure 3: The effect of **masking ratio** on (a) Something-Something V2 and (b) Kinetics-400. We take 16-frame vanilla ViT-B as default. The results show that an extremely high masking ratio (90%) achieves the best efficiency and effectiveness trade-off on both video datasets.

Figure 4: **Data efficiency** of VideoMAE representations. Our default backbone is 16-frame vanilla ViT-B. ● denotes that all models are trained for the **same** 132k iterations, and ● denotes that all models are trained for the **same** 800 epochs. Note that it takes 132k iterations to pre-train the model for 800 epochs on the full training set of Something-Something V2.

We also visualize the reconstructed examples in Appendix § E. We see that even under an extremely high masking ratio, VideoMAE can produce satisfying reconstructed results. This implies VideoMAE is able to learn useful representations that capture the holistic spatiotemporal structure in videos.

**Transfer learning: quality vs. quantity.** To further investigate the generalization ability of VideoMAE in representation learning, we transfer the learned VideoMAE from Kinetics-400 to Something-Something V2, UCF101, and HMDB51. The results are shown in Table 4, and we compare them with MoCo v3 pre-training. The models pre-trained by VideoMAE are better than those pre-trained by MoCo v3, demonstrating that our VideoMAE learns more transferable representations.

Comparing Table 2 and Table 4, the transferred representation outperforms the original VideoMAE models trained from its own dataset on UCF101 and HMDB51. In contrast, the transferred representation is worse on Something-Something V2. To figure out whether this inconsistent result is caused by the large scale of Something-Something V2, we further perform a detailed investigation by decreasing the pre-training video numbers. In this study, we run two experiments: (1) pre-training with the same epochs and (2) pre-training with the same time budget. The result is shown in Figure 4. We see that more training iterations could contribute to better performance when we decrease the size of the pre-training set. Surprisingly, even with only 42k pre-training videos, we can still obtain better accuracy than the Kinetics pre-trained models with 240k videos (68.7% vs. 68.5%). This result implies that domain shift is another important factor, and data quality is more important than data quantity in SSVP when there exists a difference between pre-training and target datasets. It also demonstrates that VideoMAE is a data-efficient learner for SSVP.

**Transfer learning: downstream action detection.** We also transfer the learned VideoMAE on Kinetics-400 to downstream action detection dataset AVA. Following the standard setting [26], we evaluate on top 60 common classes with mean Average Precision (mAP) as the metric under IoU threshold of 0.5. The results are shown in the Table 5. After self-supervised pre-training on Kinetics-400, our VideoMAE with the vanilla ViT-B can achieve 26.7 mAP on AVA, which demonstrates the strong transferability of our VideoMAE. If the pre-trained ViT-B is additionally fine-tuned on Kinetics-400 with labels, the transfer learning performance can further increase about 5 mAP (from 26.7 to 31.8). More remarkably, when we scale up the pre-training configurations with larger video datasets (e.g. Kinetics-700) or more powerful backbones (e.g. ViT-Large and ViT-Huge), VideoMAE can finally obtain better performance. For example, our ViT-L VideoMAE pre-trained on Kinetics-700 achieves 39.3 mAP and ViT-H VideoMAE pre-trained on Kinetics-400 has 39.5 mAP. These results demonstrate that the self-supervised pre-trained models transfer well not only on action classification task but on more complex action detection task.

| Method | Backbone | Pre-train Dataset | Extra Labels | $T \times \tau$ | GFLOPs | Param | mAP |
|---|---|---|:---:|:---:|:---:|:---:|:---:|
| supervised [22] | SlowFast-R101 | Kinetics-400 | ✓ | 8×8 | 138 | 53 | 23.8 |
| CVRL [53] | SlowOnly-R50 | Kinetics-400 | ✗ | 32×2 | 42 | 32 | 16.3 |
| $\rho$BYOL$_{\rho=3}$ [23] | SlowOnly-R50 | Kinetics-400 | ✗ | 8×8 | 42 | 32 | 23.4 |
| $\rho$MoCo$_{\rho=3}$ [23] | SlowOnly-R50 | Kinetics-400 | ✗ | 8×8 | 42 | 32 | 20.3 |
| MaskFeat↑312 [79] | MViT-L | Kinetics-400 | ✓ | 40×3 | 2828 | 218 | 37.5 |
| MaskFeat↑312 [79] | MViT-L | Kinetics-600 | ✓ | 40×3 | 2828 | 218 | 38.8 |
| **VideoMAE** | ViT-S | Kinetics-400 | ✗ | 16×4 | 57 | 22 | 22.5 |
| **VideoMAE** | ViT-S | Kinetics-400 | ✓ | 16×4 | 57 | 22 | 28.4 |
| **VideoMAE** | ViT-B | Kinetics-400 | ✗ | 16×4 | 180 | 87 | 26.7 |
| **VideoMAE** | ViT-B | Kinetics-400 | ✓ | 16×4 | 180 | 87 | 31.8 |
| **VideoMAE** | ViT-L | Kinetics-400 | ✗ | 16×4 | 597 | 305 | 34.3 |
| **VideoMAE** | ViT-L | Kinetics-400 | ✓ | 16×4 | 597 | 305 | 37.0 |
| **VideoMAE** | ViT-H | Kinetics-400 | ✗ | 16×4 | 1192 | 633 | **36.5** |
| **VideoMAE** | ViT-H | Kinetics-400 | ✓ | 16×4 | 1192 | 633 | **39.5** |
| **VideoMAE** | ViT-L | Kinetics-700 | ✗ | 16×4 | 597 | 305 | **36.1** |
| **VideoMAE** | ViT-L | Kinetics-700 | ✓ | 16×4 | 597 | 305 | **39.3** |

Table 5: **Comparison with the state-of-the-art methods on AVA v2.2.** All models are pre-trained and fine-tuned at image size $224^2$. We report the mean Average Precision (mAP) on validation set. "Ex. labels ✗" means only *unlabelled* data is used during the pre-training phase and the pre-trained models are directly transferred to AVA. "Ex. labels ✓" means pre-trained models are additionally fine-tuned on the pre-training dataset with *labels* before transferred to AVA. $T \times \tau$ refers to frame number and corresponding sample rate.

| Method | Backbone | Extra data | Ex. labels | Frames | GFLOPs | Param | Top-1 | Top-5 |
|---|---|---|:---:|:---:|---|:---:|:---:|:---:|
| TEINet$_{En}$ [39] | ResNet50$_{\times 2}$ | | ✓ | 8+16 | 99×10×3 | 50 | 66.5 | N/A |
| TANet$_{En}$ [40] | ResNet50$_{\times 2}$ | ImageNet-1K | ✓ | 8+16 | 99×2×3 | 51 | 66.0 | 90.1 |
| TDN$_{En}$ [74] | ResNet101$_{\times 2}$ | | ✓ | 8+16 | 198×1×3 | 88 | 69.6 | 92.2 |
| SlowFast [22] | ResNet101 | Kinetics-400 | ✓ | 8+32 | 106×1×3 | 53 | 63.1 | 87.6 |
| MViTv1 [21] | MViTv1-B | | ✓ | 64 | 455×1×3 | 37 | 67.7 | 90.9 |
| TimeSformer [6] | ViT-B | ImageNet-21K | ✓ | 8 | 196×1×3 | 121 | 59.5 | N/A |
| TimeSformer [6] | ViT-L | | ✓ | 64 | 5549×1×3 | 430 | 62.4 | N/A |
| ViViT FE [3] | ViT-L | | ✓ | 32 | 995×4×3 | N/A | 65.9 | 89.9 |
| Motionformer [50] | ViT-B | IN-21K+K400 | ✓ | 16 | 370×1×3 | 109 | 66.5 | 90.1 |
| Motionformer [50] | ViT-L | | ✓ | 32 | 1185×1×3 | 382 | 68.1 | 91.2 |
| Video Swin [38] | Swin-B | | ✓ | 32 | 321×1×3 | 88 | 69.6 | 92.7 |
| VIMPAC [64] | ViT-L | HowTo100M+DALLE | ✗ | 10 | N/A×10×3 | 307 | 68.1 | N/A |
| BEVT [76] | Swin-B | IN-1K+K400+DALLE | ✗ | 32 | 321×1×3 | 88 | 70.6 | N/A |
| MaskFeat↑312 [79] | MViT-L | Kinetics-600 | ✓ | 40 | 2828×1×3 | 218 | 75.0 | 95.0 |
| **VideoMAE** | ViT-B | Kinetics-400 | ✗ | 16 | 180×2×3 | 87 | 69.7 | 92.3 |
| **VideoMAE** | ViT-L | Kinetics-400 | ✗ | 16 | 597×2×3 | 305 | 74.0 | 94.6 |
| **VideoMAE** | ViT-S | | ✗ | 16 | 57×2×3 | 22 | 66.8 | 90.3 |
| **VideoMAE** | ViT-B | *no external data* | ✗ | 16 | 180×2×3 | 87 | 70.8 | 92.4 |
| **VideoMAE** | ViT-L | | ✗ | 16 | 597×2×3 | 305 | 74.3 | 94.6 |
| **VideoMAE** | ViT-L | | ✗ | 32 | 1436×1×3 | 305 | **75.4** | **95.2** |

Table 6: **Comparison with the state-of-the-art methods on Something-Something V2**. Our VideoMAE reconstructs normalized cube pixels and is pre-trained with a masking ratio of 90% for 2400 epochs. "Ex. labels ✗" means only *unlabelled* data is used during the pre-training phase. "N/A" indicates the numbers are not available for us.

### 4.4 Comparison with the state of the art

We compare with the previous state-of-the-art performance on the Kinetics-400 and Something-Something V2 datasets. The results are reported in Table 6 and Table 7. Our VideoMAE can easily scale up with more powerful backbones (e.g. ViT-Large and ViT-Huge) and more frames (e.g. 32). Our VideoMAE achieves the top-1 accuracy of 75.4% on Something-Something V2 and 87.4% on Kinetics-400 without using any extra data. We see that the existing state-of-the-art methods all depend on the external data for pre-training on the Something-Something V2 dataset. On the contrary, our VideoMAE without any external data significantly outperforms previous methods with the same input resolution by around 5%. Our ViT-H VideoMAE also achieves very competitive performance on the Kinetics-400 dataset without using any extra data, which is even better than ViViT-H with on

| Method | Backbone | Extra data | Ex. labels | Frames | GFLOPs | Param | Top-1 | Top-5 |
|---|---|---|---|---|---|---|---|---|
| NL I3D [77] | ResNet101 | | ✓ | 128 | 359×10×3 | 62 | 77.3 | 93.3 |
| TANet [40] | ResNet152 | ImageNet-1K | ✓ | 16 | 242×4×3 | 59 | 79.3 | 94.1 |
| TDN$_{En}$ [74] | ResNet101 | | ✓ | 8+16 | 198×10×3 | 88 | 79.4 | 94.4 |
| TimeSformer [6] | ViT-L | | ✓ | 96 | 8353×1×3 | 430 | 80.7 | 94.7 |
| ViViT FE [3] | ViT-L | ImageNet-21K | ✓ | 128 | 3980×1×3 | N/A | 81.7 | 93.8 |
| Motionformer [50] | ViT-L | | ✓ | 32 | 1185×10×3 | 382 | 80.2 | 94.8 |
| Video Swin [38] | Swin-L | | ✓ | 32 | 604×4×3 | 197 | 83.1 | 95.9 |
| ViViT FE [3] | ViT-L | JFT-300M | ✓ | 128 | 3980×1×3 | N/A | 83.5 | 94.3 |
| ViViT [3] | ViT-H | JFT-300M | ✓ | 32 | 3981×4×3 | N/A | 84.9 | 95.8 |
| VIMPAC [64] | ViT-L | HowTo100M+DALLE | ✗ | 10 | N/A×10×3 | 307 | 77.4 | N/A |
| BEVT [76] | Swin-B | IN-1K+DALLE | ✗ | 32 | 282×4×3 | 88 | 80.6 | N/A |
| MaskFeat↑352 [79] | MViT-L | Kinetics-600 | ✗ | 40 | 3790×4×3 | 218 | 87.0 | 97.4 |
| ip-CSN [68] | ResNet152 | | ✗ | 32 | 109×10×3 | 33 | 77.8 | 92.8 |
| SlowFast [22] | R101+NL | *no external data* | ✗ | 16+64 | 234×10×3 | 60 | 79.8 | 93.9 |
| MViTv1 [21] | MViTv1-B | | ✗ | 32 | 170×5×1 | 37 | 80.2 | 94.4 |
| MaskFeat [79] | MViT-L | | ✗ | 16 | 377×10×1 | 218 | 84.3 | 96.3 |
| **VideoMAE** | ViT-S | | ✗ | 16 | 57×5×3 | 22 | 79.0 | 93.8 |
| **VideoMAE** | ViT-B | *no external data* | ✗ | 16 | 180×5×3 | 87 | 81.5 | 95.1 |
| **VideoMAE** | ViT-L | | ✗ | 16 | 597×5×3 | 305 | 85.2 | 96.8 |
| **VideoMAE** | ViT-H | | ✗ | 16 | 1192×5×3 | 633 | **86.6** | **97.1** |
| **VideoMAE↑320** | ViT-L | *no external data* | ✗ | 32 | 3958×4×3 | 305 | 86.1 | 97.3 |
| **VideoMAE↑320** | ViT-H | | ✗ | 32 | 7397×4×3 | 633 | **87.4** | **97.6** |

Table 7: **Comparison with the state-of-the-art methods on Kinetics-400**. Our VideoMAE reconstructs normalized cube pixels. Here models are self-supervised pre-trained with a masking ratio of 90% for 1600 epochs on Kinetics-400. VideoMAE↑320 is initialized from its $224^2$ resolution counterpart and then fine-tuned for evaluation. "Ex. labels ✗" means only *unlabelled* data is used during the pre-training phase. "N/A" indicates the numbers are not available for us.

JFT-300M pre-training (86.6% v.s. 84.9%). When fine-tuned with larger spatial resolutions and input video frames, the performance of our ViT-H VideoMAE can further boost from 86.6% to 87.4%.

# 5 Conclusion

In this paper, we have presented a simple and data-efficient self-supervised learning method (Video-MAE) for video transformer pre-training. Our VideoMAE introduces two critical designs of extremely high masking ratio and tube masking strategy to make the video reconstruction task more challenging. This harder task would encourage VideoMAE to learn more representative features and relieve the information leakage issue. Empirical results demonstrate this simple algorithm works well for video datasets of different scales. In particular, we are able to learn effective VideoMAE only with thousands of video clips, which has significant practical value for scenarios with limited data available.

**Future work** VideoMAE could be further improved by using larger webly datasets, larger models (e.g., ViT-G) and larger spatial resolutions of input video (e.g., $384^2$). VideoMAE only leverages the RGB video stream without using additional audio or text stream. We expect that audio and text from the video data can provide more information for self-supervised pre-training.

**Broader impact** Potential negative societal impacts of VideoMAE are mainly concerned with energy consumption. The pre-training phase may lead to a large amount of carbon emission. Though the pre-training is energy-consuming, we only need to pre-train the model once. Different downstream tasks can then share the same pre-trained model via additional fine-tuning. Our VideoMAE unleashes the great potential of vanilla vision transformer for video analysis, which could increase the risk of video understanding model or its outputs being used incorrectly, such as for unauthorized surveillance.

**Acknowledgements and disclosure of funding** Thanks to Ziteng Gao, Lei Chen and Chongjian Ge for their help. This work is supported by National Natural Science Foundation of China (No. 62076119, No. 61921006), the Fundamental Research Funds for the Central Universities (No. 020214380091), Tencent AI Lab Rhino-Bird Focused Research Program (No. JR202125), and Collaborative Innovation Center of Novel Software Technology and Industrialization.

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
