# Appendix

In this appendix, we provide more details of VideoMAE from the following aspects:

- The detailed architecture illustration is in § A.
- The implementation details are in § B.
- Experimental results are in § C where there are ablation studies on the Something-Something V2 and Kinectics-400 datasets, and a downstream evaluation task (i.e., action detection). More comparisons with state-of-the-art methods on the UCF101 and HMDB51 datasets are included as well.
- Results analysis is in § D.
- Visualization of reconstructed samples is in § E.
- License of the datasets is in § F.

## A    Architectures

We use an asymmetric encoder-decoder architecture for video self-supervised pre-training and discard the decoder during the fine-tuning phase. We take the 16-frame vanilla ViT-Base for example, and the specific architectural design for the encoder and decoder is shown in Table 8. We adopt the joint space-time attention [3, 38] to better capture the high-level spatio-temporal information in the remaining tokens.

| Stage | Vision Transformer (Base) | | Output Sizes |
|---|---|---|---|
| data | stride $4{\times}1{\times}1$ on *K400* 
 stride $2{\times}1{\times}1$ on *SSV2* | | $3{\times}16{\times}224{\times}224$ |
| cube | $2{\times}16{\times}16$, $768$ 
 stride $2{\times}16{\times}16$ | | $768{\times}\mathbf{8}{\times}196$ |
| mask | tube mask 
 *mask ratio = $\rho$* | | $768{\times}\mathbf{8}{\times}[196{\times}(1\text{-}\rho)]$ |
| encoder | MHA($768$) 
 MLP($3072$) | $\times 12$ | $768{\times}\mathbf{8}{\times}[196{\times}(1\text{-}\rho)]$ |
| projector | MLP($384$) & 
 *concat learnable tokens* | | $384{\times}\mathbf{8}{\times}196$ |
| decoder | MHA($384$) 
 MLP($1536$) | $\times 4$ | $384{\times}\mathbf{8}{\times}196$ |
| projector | MLP($1536$) | | $1536{\times}\mathbf{8}{\times}196$ |
| reshape | *from $1536$ to $3{\times}2{\times}16{\times}16$* | | $3{\times}16{\times}224{\times}224$ |

Table 8: **Architectures details of VideoMAE.** We take 16-frame vanilla ViT-Base for example. "MHA" here denotes the joint space-time self-attention. The output sizes are denoted by $\{C{\times}T{\times}S\}$ for channel, temporal and spatial sizes.

## B    Implementation Details

We conduct the experiments with 64 GPUs for both pre-training and fine-tuning on the Something-Something V2 and Kinetics-400 datasets. The experiments on the smaller UCF101 and HMDB51 datasets are trained with 8 GPUs. The experiments on the AVA dataset are conducted with 32 GPUs. We linearly scale the base learning rate w.r.t. the overall batch size, *lr = base learning rate × batch size / 256*. We adopt the PyTorch [46] and DeepSpeed[2] frameworks for faster training. We have made the code[3] and pre-trained models[4] public to facilitate future research in self-supervised video pre-training.

**Something-Something V2.** Our VideoMAE is pre-trained for 800 epochs on Something-Something V2 by default. During the fine-tuning phase, we perform the uniform sampling following TSN [75]. For evaluation, all models share the same inference protocol, i.e., 2 clips × 3 crops. The default

---

[2]https://github.com/microsoft/DeepSpeed
[3]https://github.com/MCG-NJU/VideoMAE
[4]https://github.com/MCG-NJU/VideoMAE/blob/main/MODEL_ZOO.md

| config | Sth-Sth V2 | Kinetics-400 |
|---|---|---|
| optimizer | AdamW | |
| base learning rate | 1.5e-4 | |
| weight decay | 0.05 | |
| optimizer momentum | $\beta_1, \beta_2 = 0.9, 0.95$ [11] | |
| batch size | 1024 | |
| learning rate schedule | cosine decay [41] | |
| warmup epochs | 40 | |
| flip augmentation | *no* | *yes* |
| augmentation | MultiScaleCrop [75] | |

Table 9: **Pre-training setting.**

| config | Sth-Sth V2 | Kinetics-400 |
|---|---|---|
| optimizer | AdamW | |
| base learning rate | 1e-3(S), 5e-4(B,L) | 1e-3 |
| weight decay | 0.05 | |
| optimizer momentum | $\beta_1, \beta_2 = 0.9, 0.999$ | |
| batch size | 512 | 512 |
| learning rate schedule | cosine decay [41] | |
| warmup epochs | 5 | |
| training epochs | 40 (S,B), 30 (L) | 150 (S), 75 (B), 50 (L,H) |
| repeated augmentation | 2 | 2 |
| flip augmentation | *no* | *yes* |
| RandAug [15] | (9, 0.5) | (9, 0.5) |
| label smoothing [63] | 0.1 | 0.1 |
| mixup [86] | 0.8 | 0.8 |
| cutmix [85] | 1.0 | 1.0 |
| drop path | 0.1 (S,B), 0.2 (L,H) | |
| dropout | 0.5 (L) | 0.5 (L,H) |
| layer-wise lr decay [4] | 0.7 (S),0.75 (B,L) | 0.7 (S),0.75 (B,L,H) |

Table 10: **End-to-end fine-tuning setting in Table 6 and Table 7.**

| config | Sth-Sth V2 |
|---|---|
| optimizer | SGD |
| base learning rate | 0.1 |
| weight decay | 0 |
| optimizer momentum | 0.9 |
| batch size | 1024 |
| learning rate schedule | cosine decay |
| warmup epochs | 10 |
| training epochs | 100 |
| augmentation | MultiScaleCrop |

Table 11: **Linear probing setting.**

settings of pre-training, fine-tuning, and linear probing are shown in Table 9, Table 12, and Table 11. For supervised training, we follow the recipe in [21] and train from scratch for 100 epochs. Note that we use *no flip augmentation* during both the pre-training and fine-tuning phase. We additionally adopt the repeated augmentation [31] during the fine-tuning phase in Table 6, which can further increase the Top-1 accuracy by 0.1% - 0.3%.

**Kinetics-400.** Our VideoMAE is pre-trained for 800 epochs on Kinetics-400 by default. During the fine-tuning phase, we perform the dense sampling following Slowfast [22]. For evaluation, all models share the same inference protocol, i.e., 5 clips × 3 crops. The default settings of pre-training and fine-tuning are shown in Table 9 and Table 12. For supervised training from scratch, we follow the recipe in [21] and train the model for 200 epochs. Note that we adopt the repeated augmentation [31] during the fine-tuning phase in Table 7, which can further increase the Top-1 accuracy by 0.8% - 1.0%.

| config | AVA v2.2 | |
|---|---|---|
| additional fine-tuning on Kinetics | ✗ | ✓ |
| optimizer | AdamW | |
| base learning rate | 1e-3 (S), 2.5e-4 (B, L), 5e-4 (H) | 5e-4 |
| weight decay | 0.05 | |
| optimizer momentum | $\beta_1, \beta_2 = 0.9, 0.999$ | |
| batch size | 128 | |
| learning rate schedule | cosine decay [41] | |
| warmup epochs | 5 | |
| training epochs | 30 (S, B, L), 20 (H) | 30 (S, B) 20 (L, H) |
| repeated augmentation | no | |
| flip augmentation | yes | |
| drop path | 0.2 | |
| layer-wise lr decay [4] | 0.6 (S), 0.75 (B, L), 0.8 (H) | 0.6 (S), 0.75 (B), 0.8 (L, H) |

Table 12: **End-to-end fine-tuning setting in Table 5.**

**UCF101.** We follow a similar recipe on Kinetics for pre-training. Our VideoMAE is pre-trained with a masking ratio of 75% for 3200 epochs. The batch size and base learning rate are set to 192 and 3e-4, respectively. Here, 16 frames with a temporal stride of 4 are sampled. For fine-tuning, the model is trained with repeated augmentation [31] and a batch size of 128 for 100 epochs. The base learning rate, layer decay and drop path are set to 5e-4, 0.7 and 0.2, respectively. For evaluation, we adopt the inference protocol of 5 clips × 3 crops.

**HMDB51.** Our VideoMAE is pre-trained with a masking ratio of 75% for 4800 epochs. The batch size and base learning rate are set to 192 and 3e-4, respectively. Here, 16 frames with a temporal stride of 2 are sampled. For fine-tuning, the model is trained with repeated augmentation [31] and a batch size of 128 for 50 epochs. The base learning rate, layer decay and drop path are set to 1e-3, 0.7 and 0.2, respectively. For evaluation, we adopt the inference protocol of 10 clips × 3 crops.

**AVA.** We follow the action detection architecture in Slowfast [22] and use the detected person boxes from AIA [65]. The default settings of fine-tuning are shown in Table 12. For data augmentations, we resize the short side of the input frames to 256 pixels. We apply a random crop of the input frames to 224×224 pixels and random flip during training. We use only ground-truth person boxes for training and the detected boxes with confidence ≥0.8 for inference.

## C   Additional Results

### C.1   Training schedule

Figure 5 shows the influence of the longer pre-training schedule on the Something-Something V2 and Kinetics-400 datasets. We find that a longer pre-training schedule brings slight gains to both datasets. In the main paper, our VideoMAE is pre-trained for 800 epochs by default.

### C.2   Comparison with the state-of-the-art methods

We present the detailed comparison with the state-of-the-art on UCF101 and HMDB51 in Table 13. Figure 6 additionally shows that our VideoMAE is a data-efficient learner that allows us to effectively train video transformers only from limited video data (e.g., 9.5k clips in UCF101, and 3.5k clips in HMDB51) without any ImageNet pre-training. VideoMAE significantly outperforms training from scratch, MoCo v3 pre-training [14], and the previous best performance from Vi$^2$CLR [18] without extra data on these small-scale video datasets. Compared with those large-scale video datasets, these two small datasets are more proper to verify the effectiveness of VideoMAE, as training large ViT models is more challenging on small datasets.

## D   Model result analysis

In this section, we add the analysis of model results. As shown in Figure 7 and Figure 8, our VideoMAE bring significant gain for most categories on SSV2, which implies that our VideoMAE

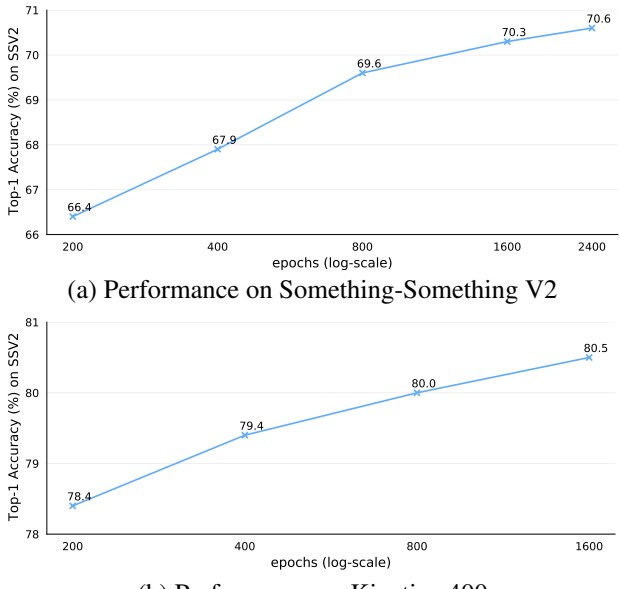

(a) Performance on Something-Something V2

(b) Performance on Kinetics-400

Figure 5: The effect of **training schedules** on (a) Something-Something V2 and (b) Kinetics-400. Here each point is a full training schedule. Our default ViT-B backbone is described in Table 8.

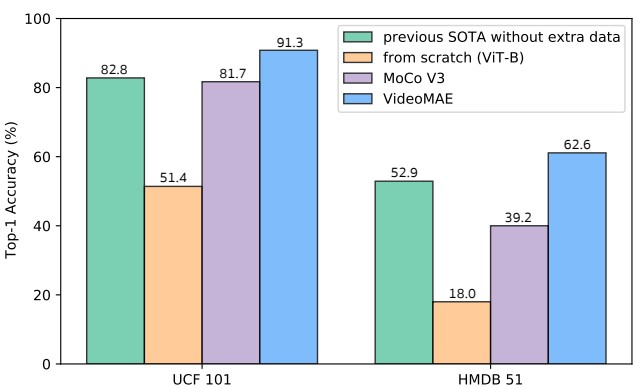

Figure 6: Comparison with VideoMAE, MoCo v3 [14], and Vi$^2$CLR [18] on UCF101 and HMDB51.

can capture more spatiotemporal structure representations than ImageMAE and ImageNet-21k supervised pre-trained model. On the other hand, we also notice that our VideoMAE performs slightly worse than other two models on some categories. To better understand how the model works, we select several examples from validation set. The examples are shown in Figure 9. For the example in the 1st row, we find our VideoMAE might not capture the motion information from very small object. We suspect that tokens containing the small motion might all be masked due to our extremely high masking ratio, so our VideoMAE could hardly reconstruct the masked small motion pattern. For the example in the 2nd row, we find our VideoMAE could capture the deformation of objects and movement from the squeeze of the hand, while this cannot be discriminated by image pre-training. We leave more detailed analysis of our VideoMAE for future work.

## E   Visualization

We show several examples of reconstruction in Figure 10 and Figure 11. Videos are all randomly chosen from the validation set. We can see that even under an extremely high masking ratio, VideoMAE can produce satisfying reconstructed results. These examples imply that our VideoMAE

| Method | Backbone | Extra data | Frames | Param | Modality | UCF101 | HMDB51 |
|---|---|---|---|---|---|---|---|
| OPN [35] | VGG | UCF101 | N/A | N/A | V | 59.6 | 23.8 |
| VCOP [82] | R(2+1)D | UCF101 | N/A | N/A | V | 72.4 | 30.9 |
| CoCLR [29] | S3D-G | UCF101 | 32 | 9M | V | 81.4 | 52.1 |
| Vi$^2$CLR [18] | S3D | UCF101 | 32 | 9M | V | 82.8 | 52.9 |
| **VideoMAE** | ViT-B | *no external data* | 16 | 87M | V | **91.3** | **62.6** |
| SpeedNet [5] | S3D-G | Kinetics-400 | 64 | 9M | V | 81.1 | 48.8 |
| VTHCL [84] | SlowOnly-R50 | Kinetics-400 | 8 | 32M | V | 82.1 | 49.2 |
| Pace [73] | R(2+1)D | Kinetics-400 | 16 | 15M | V | 77.1 | 36.6 |
| MemDPC [28] | R-2D3D | Kinetics-400 | 40 | 32M | V | 86.1 | 54.5 |
| CoCLR [29] | S3D-G | Kinetics-400 | 32 | 9M | V | 87.9 | 54.6 |
| RSPNet [12] | S3D-G | Kinetics-400 | 64 | 9M | V | 93.7 | 64.7 |
| VideoMoCo [45] | R(2+1)D | Kinetics-400 | 16 | 15M | V | 78.7 | 49.2 |
| Vi$^2$CLR [18] | S3D | Kinetics-400 | 32 | 9M | V | 89.1 | 55.7 |
| CVRL [53] | SlowOnly-R50 | Kinetics-400 | 32 | 32M | V | 92.9 | 67.9 |
| CVRL [53] | SlowOnly-R50 | Kinetics-600 | 32 | 32M | V | 93.6 | 69.4 |
| CVRL [53] | Slow-R152 (2×) | Kinetics-600 | 32 | 328M | V | 94.4 | 70.6 |
| CORP$_f$ [32] | SlowOnly-R50 | Kinetics-400 | 32 | 32M | V | 93.5 | 68.0 |
| $\rho$SimCLR$_{\rho=2}$ [23] | SlowOnly-R50 | Kinetics-400 | 8 | 32M | V | 88.9 | N/A |
| $\rho$SwAV$_{\rho=2}$ [23] | SlowOnly-R50 | Kinetics-400 | 8 | 32M | V | 87.3 | N/A |
| $\rho$MoCo$_{\rho=2}$ [23] | SlowOnly-R50 | Kinetics-400 | 8 | 32M | V | 91.0 | N/A |
| $\rho$BYOL$_{\rho=2}$ [23] | SlowOnly-R50 | Kinetics-400 | 8 | 32M | V | 92.7 | N/A |
| $\rho$BYOL$_{\rho=4}$ [23] | SlowOnly-R50 | Kinetics-400 | 8 | 32M | V | 94.2 | 72.1 |
| MIL-NCE [43] | S3D | HowTo100M | 32 | 9M | V+T | 91.3 | 61.0 |
| MMV [1] | S3D-G | AS+HTM | 32 | 9M | V+A+T | 92.5 | 69.6 |
| CPD [36] | ResNet50 | IG300k | 16 | N/A | V+T | 92.8 | 63.8 |
| ELO [51] | R(2+1)D | Youtube8M-2 | N/A | N/A | V+A | 93.8 | 67.4 |
| XDC [2] | R(2+1)D | Kinetics-400 | 32 | 15M | V+A | 84.2 | 47.1 |
| XDC [2] | R(2+1)D | IG65M | 32 | 15M | V+A | 94.2 | 67.1 |
| GDT [49] | R(2+1)D | Kinetics-400 | 32 | 15M | V+A | 89.3 | 60.0 |
| GDT [49] | R(2+1)D | IG65M | 32 | 15M | V+A | 95.2 | 72.8 |
| **VideoMAE** | ViT-B | Kinetics-400 | 16 | 87M | V | **96.1** | **73.3** |

Table 13: **Comparison with the state-of-the-art methods on UCF101 and HMDB51.** Our VideoMAE reconstructs normalized cube pixels and is pre-trained with a masking ratio of 75% for 3200 epochs on UCF101 and 4800 epochs on HMDB51, respectively. We report fine-tuning accuracy for evaluation. 'V' refers to visual only, 'A' is audio, 'T' is text narration. "N/A" indicates the numbers are not available for us.

is able to learn more representative features that capture the holistic spatiotemporal structure in videos.

# F   License of Data

All the datasets we used are commonly used datasets for academic purpose. The license of the Something-Something V2[5] and UCF101[6] datasets is custom. The license of the Kinetics-400[7], HMDB51[8] and AVA[9] datasets is CC BY-NC 4.0[10].

---

[5]URL: https://developer.qualcomm.com/software/ai-datasets/something-something
[6]URL: https://www.crcv.ucf.edu/data/UCF101.php
[7]URL: https://www.deepmind.com/open-source/kinetics
[8]URL: https://serre-lab.clps.brown.edu/resource/hmdb-a-large-human-motion-database
[9]URL: https://research.google.com/ava/index.html
[10]URL: https://creativecommons.org/licenses/by/4.0

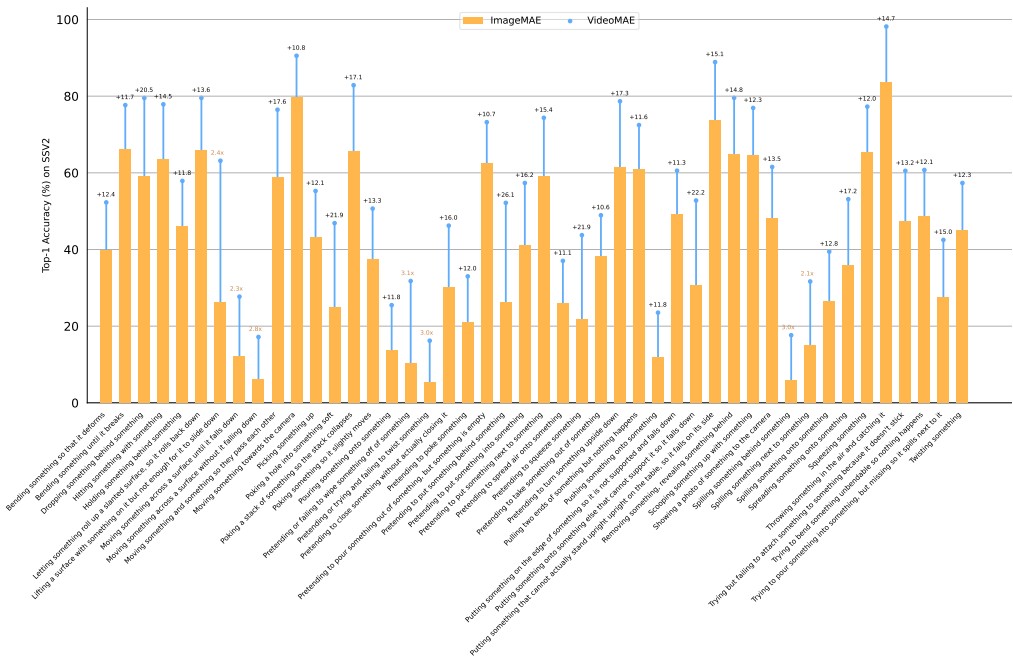

(a) Categories that VideoMAE outperforms ImageMAE. We only show those gain larger than 10%.

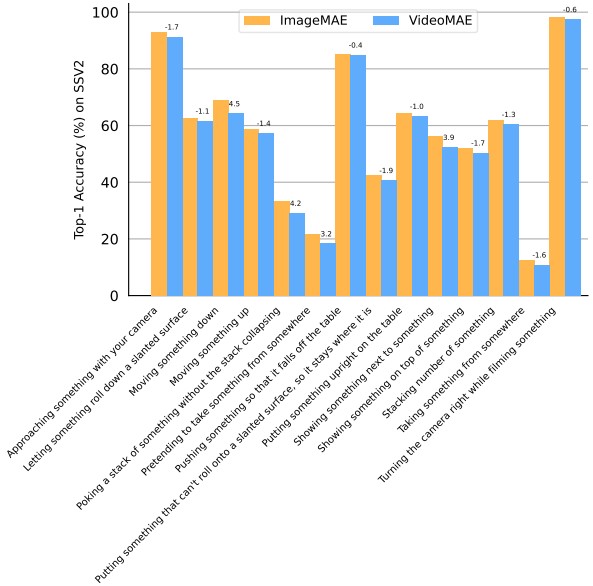

(b) Categories that ImageMAE outperforms VideoMAE.

Figure 7: ImageMAE (64.8%) vs. VideoMAE (69.6%) on Something-Something V2.

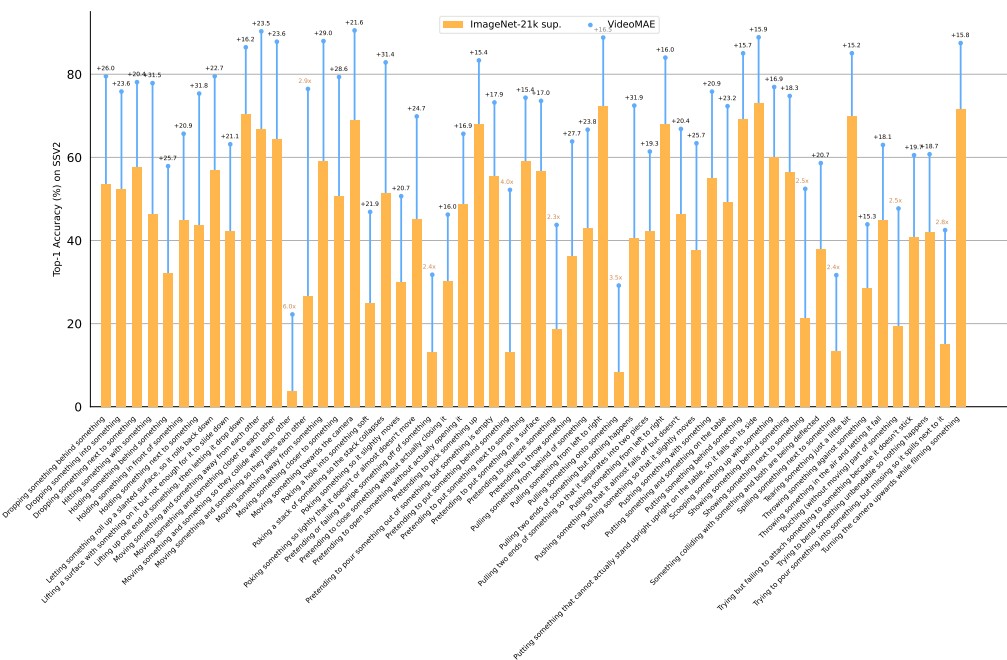

(a) Categories that VideoMAE outperforms ImageNet-21k supervised pre-trained model. We only show those gain larger than 15%.

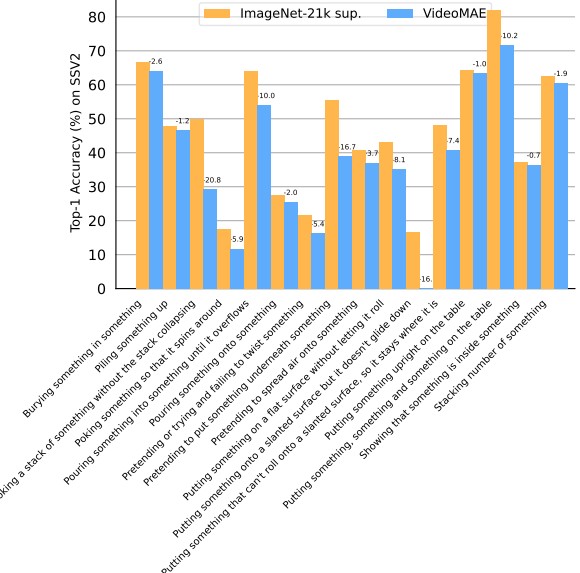

(b) Categories that ImageNet-21k supervised pre-trained model outperforms VideoMAE.

Figure 8: ImageNet-21k supervised pre-trained model (61.8%) vs. VideoMAE (69.6%) on Something-Something V2.

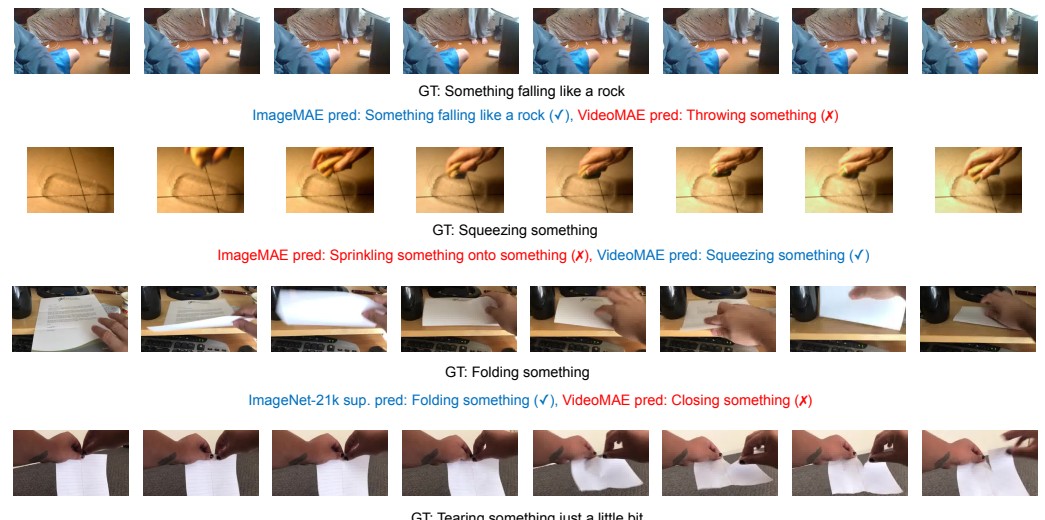

Figure 9: Prediction examples of different models on Something-Something V2. For each example drawn from the validation dataset, the predictions with blue text indicating a correct prediction and red indicating an incorrect one. "GT" indicates the ground truth of the example.

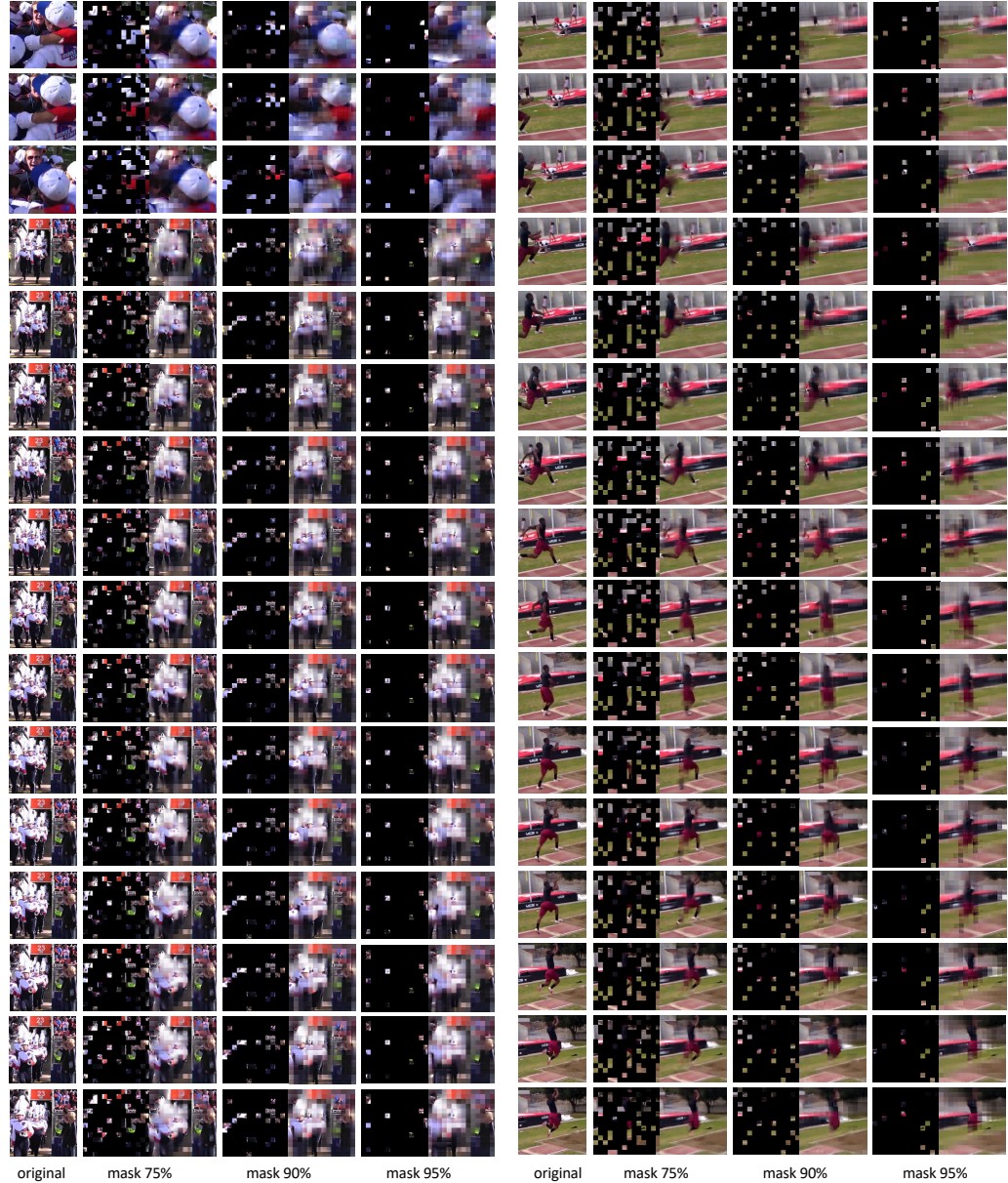

original    mask 75%    mask 90%    mask 95%      original    mask 75%    mask 90%    mask 95%

Figure 10: **Uncurated random videos** on Kinetics-400 *validation* set. We show the original video squence and reconstructions with different masking ratios. Reconstructions of videos are predicted by our VideoMAE pre-trained with a masking ratio of 90%.

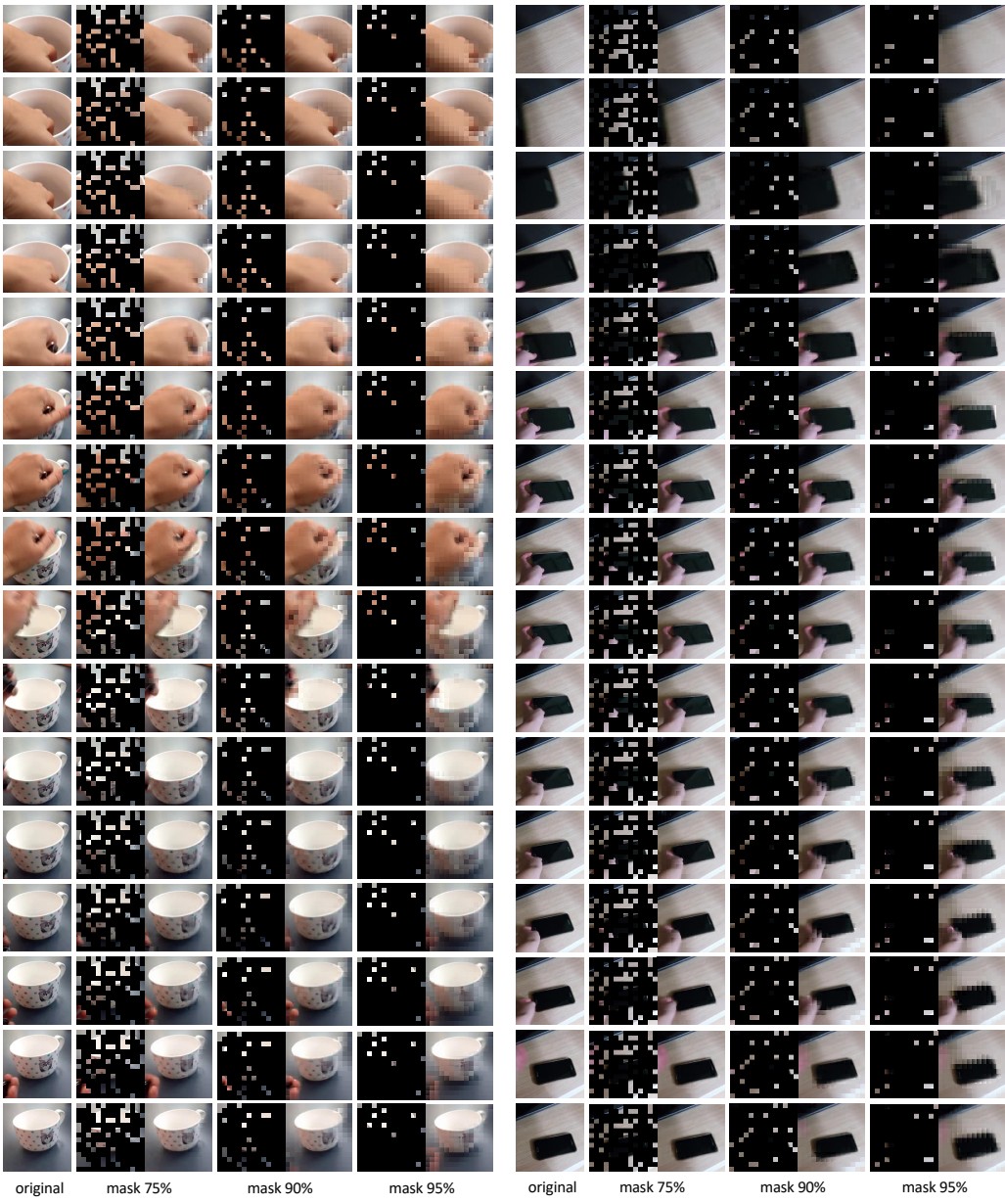

Figure 11: **Uncurated random videos** on Something-Something V2 *validation* set. We show the original video squence and reconstructions with different masking ratios. Reconstructions of videos are all predicted by our VideoMAE pre-trained with a masking ratio of 90%.