# OpenReview forum: "VideoMAE: Masked Autoencoders are Data-Efficient Learners for Self-Supervised Video Pre-Training"
_NeurIPS.cc/2022/Conference — NeurIPS 2022 Accept_

### Official Review · Reviewer_bvLA · 2022-06-20

**Rating:** 6
**Confidence:** 4
**Soundness:** 3 good
**Presentation:** 3 good
**Contribution:** 2 fair

**Summary:**

The paper presents a self-supervised learning method to learn representations of videos. The method is based on being able to decode masked elements from an encoded video, similar to the masked autoencoder approaches to image representation learning. The authors perform extensive experiments on different video classification datasets and ablation studies over the different design choices. As a result, the model obtains state-of-the-art performance and is able to perform well on smaller datasets without needing additional data, which is not common for self-supervised methods.

**Questions:**

- How do the representations learned on one dataset transfer to other equally large scale datasets? Is there a best way to apply the model beyond training from scratch on a particular dataset? If so, how can we determine it?
- What kind of mistakes does the model do? Is there a pattern there that could help us understand better how the model works?

**Limitations:**

The discussion on limitations and potential negative societal impacts of the submission is quite limited. However, there are no major ethical considerations with video representation learning in general. The discussion could be extended by mentioning potential negative applications when applying the method to sensitive data.

**Strengths And Weaknesses:**

**Strengths:**
- Comprehensive ablation studies that help understand which design choices work best
- Very strong results on multiple datasets, especially on smaller scale datasets without using additional data
- Simple idea that should be reproducible with the information in the paper and the submitted code

**Weaknesses:**
- The work is incremental and mostly consists in extending Masked AutoEncoders to videos. The small design choices required to adapt ViT methods for images to videos (such as temporal striding, temporal size of the input tokens, etc.) are well understood and the idea that the same spatial regions are correlated over time are well known in the video literature.
- The analysis of the model result is shallow and limited to showing the performance of the model with a certain configuration, without analyzing its mistakes.

**Detailed comment:**

This paper extends the idea of Masked AutoEncoders to video and runs multiple benchmarks to understand the effect of different design choices. While the paper is relatively incremental, there is a significant engineering effort and the authors are able to obtain strong results on multiple video classification benchmarks. One important insight is that this model is able to perform well on small datasets, contrary to other self-supervised learning methods. However, the analysis of the strengths and weaknesses of the model is quite limited.

Overall it is a good submission with a simple idea and impactful experimental results, therefore I argue for its acceptance.

---

> ### Author Response · Authors · 2022-08-02
> **Response to Reviewer bvLA**
>
> ***1. How do the representations learned on one dataset transfer to other equally large scale datasets?***
>
> Thanks for your question. We have explored the feature transferability of our VideoMAE in Table 4. In this setting, the ViT models are initialized from the self-supervised pre-trained VideoMAE on Kinetics-400 datasets. We then fine-tune the model on target datasets via end-to-end training. In addition to this paradigm, we can also train a linear classifier on top of the frozen backbone with standard cross-entropy loss. This protocol requires very little computational cost, but may lead to relatively worse performance. Besides, partial fine-tuning is another alternative, and make a trade-off between the above two methods.
>
> ***2. Is there a best way to apply the model beyond training from scratch on a particular dataset?***
>
> Thanks for your question. Although we have verified the ability of our VideoMAE to train a video transformer on the video dataset itself without using any extra data, we also surprisingly find our VideoMAE has strong feature transferability when scaling up the pre-training data. We try to pre-train our VideoMAE (i.e., ViT-L 16x2) on a larger Kinetics-700 dataset with 800 epochs and then transfer the representation to SSV2 dataset. This model can achieve 75.0 Top-1 accuracy on SSV2, which is superior to the one directly pre-trained on SSV2 (75.0 vs. 74.2). How to apply the VideoMAE pretrained models beyond simple fine-tuning is another important research direction, which is out of the scope of this paper. In the future, we can improve this transfer learning strategy by employing more advaned technique such as learning without forgetting or co-finetuning.
>
> ***3. What kind of mistakes does the model do? Is there a pattern there that could help us understand better how the model works***
>
> - We thank for the suggestion and have added the analysis of model results in our paper. Please see Figure 8, Figure 9 and Figure 10 in the supplementary files. As shown in Figure 8 and Figure 9, our VideoMAE brings significant gains for most categories on SSV2, which implies that our VideoMAE can capture more spatiotemporal structure representations than ImageMAE and ImageNet-21k supervised pre-trained model. On the other hand, we also notice that our VideoMAE performs slightly worse than other two models on some categories.
>
> - To better understand how the model works, we select several examples from validation set. The examples are shown in Figure 10. For the example in the 1st row, we find our VideoMAE might not capture the motion information from a very small object. We suspect that tokens containing the small motion might all be masked due to our extremely high masking ratio, so our VideoMAE could hardly reconstruct the masked small motion pattern. For the example in the 2nd row, we find our VideoMAE could capture the deformation of objects and movement from the squeeze of the hand, while this cannot be discriminated by image pre-training. We leave a more detailed analysis of our VideoMAE for future work.
>
> ***4. Potential negative applications.***
>
> We thank for the suggestion and have added in the paper that our VideoMAE may contain representations learned from the sensitive data. These representations may reveal the data properties of the sensitive data.

---

### Official Review · Reviewer_WGm9 · 2022-06-29

**Rating:** 7
**Confidence:** 5
**Soundness:** 3 good
**Presentation:** 4 excellent
**Contribution:** 3 good

**Summary:**

This paper studies the Masked AutoEncoder in the context of video self-supervised learning. It adapts the MAE technique to the video domain and propose tube masking along with a high masking ratio for videos. The experiments are conducted on a set of standard video action recognition tasks, showing the effectiveness of the approach. In addition, with this model, the authors also find that VideoMAE is data efficient — when sufficiently trained, the model achieves strong performance with a fraction of the training data. The main concerns of this paper are novelty, lack of comparison to a similar prior work and insufficient ablation studies.  Overall, if these concerns are properly address, I am happy to see this work accepted.

**Questions:**

L255-258, and Table 1c, the lower SSv2 performance of 16x1 compared to 16x2 I believe is mainly due to the shorter temporal support of 16x1 model (~1.33secs vs. 2.67secs for 12FPS SSv2 videos). It is quite well-known SSv2 dataset requires temporal modeling and shorter temporal support will hurt the model results (see [2] TimeSformer). In my view, this has less to do with the reconstruction target during pre-training. A better experiment will be using 16x1, and 16x2 to pre-train the model, and then fine-tune with the same 16x2 config, this removes the difference caused by temporal support. Though there is a slight input mismatch between pre-train and fine-tune for 16x1 model. I also encourage the authors to find alternatives setups to compare them fairly.

**Ethics Review Area:**

["Inadequate Data and Algorithm Evaluation"]

**Limitations:**

I don't see potential negative societal impact being discussed in this work.

**Strengths And Weaknesses:**

Strengths:

- The proposed model is simple yet quite effective. As the core task is video classification, it may benefit a lot of down-stream video tasks as a “foundation” model.
- This work shows that VideoMAE is very data efficient, this seems to be not studied in the image MAE paper. Specifically, the authors show that, when sufficiently trained, the model achieves strong performance with a fraction of the training data. Similarily, they also show the model performs much better than alternative approaches on small-scale dataset such as HMDB51.

Weaknesses:

- The proposed Tube Masking has been studied in VIMPAC [1] with a name “block masking”, this undermines the novelty of the approach. I suggest the authors cite/discuss and compare with this work.
- The novelty is limited in the sense that the work seems to simply combine MAE modeling techniques with the tube masking in VIMPAC with additional minor adjustments, e.g., increase masking ratio of tube masking. But please note, I only put novelty as a minor concern, as I do value the engineering effort of this paper and its potential benefits to the video understanding community.
- The ablation study is conducted on a single dataset SSv2. However, as shown in TimeSformer [2], the conclusion may vary a lot when looking at a different dataset, e.g., Kinetics-400 (K400). As SSv2 requires more temporal modeling while K400 videos are mostly stationary. I would imagine many conclusions will change. I strongly suggest the authors add additional ablation results on K400. I also commented in the question section about a specific ablation study where I believe the dataset property may affect the conclusion a lot.

[1] Tan, H., Lei, J., Wolf, T. and Bansal, M., 2021. Vimpac: Video pre-training via masked token prediction and contrastive learning. *arXiv preprint arXiv:2106.11250*

[2]Bertasius, G., Wang, H. and Torresani, L., 2021, July. Is space-time attention all you need for video understanding?. In *ICML.*

---

> ### Author Response · Authors · 2022-08-02
> **Response to Reviewer WGm9**
>
> ***1. Cite, discuss, and compare with VIMPAC***
>
> Thanks for pointing out VIMPAC. We have added the citation of this work in our related work section. Also, we have included its results in **Table 5** and **Table 6** in our revised manuscript, and **Table 13** and **Table 14** in our supplementary files.
>
> We would like to summarize the key differences between VIMPAC and our VideoMAE from three aspects: 1) Masking strategy. Block masking in VIMPAC constructs the 3D-contiguous masking cube, while tube masking in our VideoMAE constructs the 1D-contiguous masking cube. 2) Input data. VIMPAC uses a pre-trained VQ-VAE to discrete the input video into discrete video tokens, while our VideoMAE uses joint space-time cube embedding to obtain 3D tokens. 3) Prediction target. VIMPAC performs pre-training by predicting the discrete tokens from pre-trained VQ-VAE and introducing a contrastive learning head. Our VideoMAE performs pre-training by directly predicting the normalized cube pixels, which is simpler and more effective.  These discussions have been added in the related work section.
>
> ***2. Ablation studies on the Kinetics-400 dataset***
>
> Thanks for your valuable suggestion. We strongly agree that some conclusions might change in a different dataset. We have shown the effect of the masking ratio and training schedule in Figure 3 and Figure 6. As suggested, we additionally conduct most of ablation studies on Kinetics-400, including mask sampling strategy, reconstruction target, pre-training strategy and pre-training dataset. The results are all shown in Table 12 of our supplementary files. We analyze and summarize the results below:
> - **Masking strategy.** The conclusion of these experimental results is in accord with one on Something-Something V2 as the tube masking with the high masking ratio (i.e., 90\%) also achieves better performance than one with the lower masking ratio or other masking strategies. One may note that the performance gap is lower than one on Something-Something V2. We argue that the Kinetics videos are mostly stationary and scene-related. The effect of temporal modeling is not obvious.
> - **Reconstruction target.** We find that the reconstruction of 32 frames with stride 2 gives the nearly same performance as the same input and target setting (79.5\% vs. 79.4\%). For simplicity, we still use the same input and target setting for other experiments on Kinetics-400.
> - **Pre-training strategy.** The conclusion from these results is the same as one on Something-Something V2, where our VideoMAE pre-training outperforms supervised training or training from scratch.
> - **Pre-training dataset.** We note that domain shift between pre-training and target datasets can still be an issue for a bigger dataset, Kinetics-400, as the same for Something-Something V2.
>
> ***3. Fine-tuning configurations on SSV2***
>
> Thanks for your question. We strongly agree that SSV2 dataset requires temporal modeling and shorter temporal support will hurt the model results. Therefore, we perform the **uniform sampling** following TSN[49] during the fine-tuning phase of SSV2, which is described in Appendix B. Implementation Details. We have made the fine-tuning setting clearer in the revised version. As suggested, we also conducted additional experiments on SSV2. The results are shown in Table 13 of our revised supplementary files. We find that stride sampling, e.g., 16x2, on SSV2 during fine-tuning is not optimal. For multi-clip testing, 16x2 needs more testing views to achieve similar performance with TSN sampling. For efficiency and fair comparison, we adopt TSN sampling as our default fine-tuning protocol on SSV2. We leave a more detailed analysis of temporal support for future work.

---

> > ### Comment · Reviewer_WGm9 · 2022-08-08
> > **Response to authors**
> >
> > Thanks the authors for the details response. Most of my concerns are addresses. I would consider raise my rating in my final review.

---

### Official Review · Reviewer_C915 · 2022-07-09

**Rating:** 5
**Confidence:** 5
**Soundness:** 3 good
**Presentation:** 3 good
**Contribution:** 2 fair

**Summary:**

This paper presents a self-supervised video representation learning method (VideoMAE) based on transformer design. The idea of this approach is inspired by the ImageMAE [22] which is based on the masking and reconstruction strategy. In VideoMAE, applying a high masking ratio and tube masking strategy lead to a challenging video reconstruction task which is the key to learn more representative features. Empirical results demonstrate the effectiveness of approach on 4 different video datasets in different settings.


**Questions:**

1. Regarding weakness 1 and 2, I would like authors to clearly state the contribution of this work to video SSL and the important differences to previous approaches.
2.  Regarding weakness 3, I would like authors to clarify the reason behind stating that method [53] uses extra labels during pre-training.
3. Regarding weakness 4, I would like to see the results of VideoMAE on SSV2 when pre-trained on K400 with the model architecture and training setting of the final best model. This allows us to have a fair comparison to previous methods in the literature that use only K400 for pre-training.

**Limitations:**

Yes

**Strengths And Weaknesses:**

Strength:
- The paper is well-written and easy to follow.
- The motivation and the method are clearly elaborated.
- The experiments support most of the claims in the paper.
- The method outperforms existing approaches on challenging video benchmarks.


Weaknesses:
1. In my opinion, the claim of the first contribution (line 64) needs to be tone down as we have multiple works on the video SSL based on the idea of masking and predicting such as [53,50], in which the results are almost on par with the proposed approach. While there are some differences between the proposed approach and the existing ones which has been also mentioned by the authors in the submission, the claim of "the first masked video autoencoder that performs well for SSVP on small-scale video dataset" is not entirely precise.
2. While the proposed method presents an approach that performs on-par with SOTA, the novelty of videoMAE remains limited. In fact, (a) the idea of masking and reconstruction for video SSL has been explored before [53,50]. (b) the masking strategy (tube masking) has been already explored in [53] (c ) the idea of passing non-masked tokens to the transformer encoder which results in an efficient encoding has been explored in ImageMAE[22] paper.
3. In Table 5 of the main paper and Table 13 of supp, MaskFeat[53] results on SSV2 shown "supervised" labels, while based on [53], the pre-training is done on K400 and K600 unlabaled data and then they have a fine-tuning step on SSV2. It would be good if the authors can clarify this. To me, [53] follows the standard SSL paradigm for these downstream tasks.
4. While the feature transferability is investigated in Table 4, as far as I understood, the main set of experiments of comparison to the SOTA in Table 5 for SSV2 dataset is based on the pre-training step on SSV2. However, in the most of the previous works, the common practice is to consider a single pre-trained model which has been trained on a large-scale dataset such as K400/K600 and then evaluate on downstream tasks such as SSV2. I'm curious to see the result of VideoMAE on SSV2 when pre-trained on K400 with the same design and training setting as in Table 5 which makes the comparison of feature transferability comparable to SOTA.

---

> ### Author Response · Authors · 2022-08-02
> **Response to Reviewer C915**
>
> ***1. The claim of the first contribution is not entirely precise and needs to be toned down.***
>
> Thanks for the comments. We agree that our VideoMAE shares similar motivation of exploring masking visual modeling for self-supervised video pre-training with BEVT [50] and MaskFeat [53]. We also discussed the difference of VideoMAE with these methods in the related work section. As suggested, we have toned down the claim in the revised version to avoid ambiguity for readers: `We present a simple but effective video masked autoencoder that unleashes the potential of vanilla vision transformer for video data and performs well for SSVP on relatively small-scale video datasets.`
>
> ***2. While the proposed method presents an approach that performs on par with SOTA, the novelty of videoMAE remains limited.***
>
> We agree that our VideoMAE shares some similar spirits (e.g., masking and reconstruction) with existing methods [50,53] and imageMAE [22]. The contribution of our VideoMAE resides in how we effectively deploy these spirits and make novel discoveries on video data. The BEVT [50] reconstructs discrete tokens from the pre-trained VQ-VAE, and MaskFeat [53] reconstructs from the HOG features. Different from these methods, our VideoMAE directly reconstructs the original video data via vanilla ViTs and discovers three unnoticed but important findings, as illustrated in L09-16, which have not been explored in existing studies. With simplicity and effectiveness, our VideoMAE serves as a strong baseline to benefit future research in this area.
>
> ***3. Clarify the reason behind stating that MaskFeat [53] uses extra labels during pre-training.***
>
> As described in Appendix B.1 in MaskFeat [53], `For extra-large, long-term video models with 312 and 352 spatial resolutions as well as 32×3 and 40×3 temporal durations, we initialize from their 224 resolution, 16×4 duration counterparts ……`, This illustration indicates that under the Kinectics video dataset, when input frames are more than usual or in a higher resolution, MaskFeat [53] does not pre-train the model with longer and larger inputs (e.g., MViT-L) via self-supervision. Instead, MaskFeat directly fine-tunes these larger models from the standard 224*224 models via label supervision. As shown in Appendix B.3 in MaskFeat [53]: `We fine-tune the pre-trained MViT-L↑312, 40×3 Kinetics models ……`. This indicates that the MViT-L↑312, 40×3 Kinetics models utilize extra labels from the Kinetics dataset. We also contact the authors of MaskFeat and verify that the above illustration is correct.
>
> ***4. VideoMAE on SSV2 when pre-trained on K400***
>
> Thanks for this suggestion. We have conducted evaluations on SSV2 and the results are included in **Table 5 of the revised manuscript**. The ViT models are initialized from the self-supervised pre-trained VideoMAE on Kinetics-400 datasets.
> |       Method        |  Backbone  | Pre-training Data | Extra Label | Top-1 | Top-5 |
> | :-----------------: | :--------: | :---------------: | :---------: | :---: | :---: |
> | VideoMAE (original) | ViT-B 16x4 |     ***no***      |   &cross;   | 70.6  | 92.7  |
> |   VideoMAE (new)    | ViT-B 16x4 |   Kinetics-400    |   &cross;   | 69.7  | 92.3  |
> | VideoMAE (original) | ViT-L 16x4 |     ***no***      |   &cross;   | 74.2  | 94.7  |
> |   VideoMAE (new)    | ViT-L 16x4 |   Kinetics-400    |   &cross;   | 74.0  | 94.6  |
>
> As shown in the table, our VideoMAE can also transfer well on SSV2 when pre-trained on Kinetics-400. The results greatly surpass previous ViT-based models pre-trained on Kinetics-400, such as ViViT (74.0 v.s 65.9) and Motionformer (74.0 v.s 68.1). However, the representations from VideoMAE are still slightly worse than that directly pre-trained on SSV2. We have discussed this observation in L286-L296 that domain shift is another important factor for feature transferability.

---

> > ### Comment · Reviewer_C915 · 2022-08-05
> > **Response to authors**
> >
> > I thank the authors for their effort in preparing the rebuttal. While I still have not fully convinced about the extent of the novelty of this approach, most of my other concerns regarding the emprical evaluations have been addressed and I will consider that in my final decision.

---

### Author Response · Authors · 2022-08-02
**General Response to all reviewers**

We thank all reviewers' efforts in reviewing our paper and giving insightful comments and valuable suggestions. We are glad to find that reviewers generally acknowledge the following simplicity, effectiveness, and data-efficiency in the following:

- **Simplicity and effectiveness.** The method outperforms existing approaches on challenging video benchmarks [C915]. The proposed model is simple yet effective. As the core task is video classification, it may benefit a lot of downstream video tasks as a 'foundation' model [WGm9]. I do value the engineering effort of this paper and its potential benefits to the video understanding community. [WGm9]

- **Data efficiency.** This work shows that VideoMAE is very data efficient, which seems to be not studied in the image MAE paper. Specifically, the authors show that, when sufficiently trained, the model achieves strong performance with a fraction of the training data.  [WGm9]. Very strong results on multiple datasets, especially on smaller scale datasets without using additional data [bvLA].

As suggested by the reviewers, we have included the following contents in our revised manuscript and supplementary files for improvement. Our modifications are marked in blue. We summarize the major revisions as follows. The detailed responses can be found in the following response sections to the reviewers.

- **Extended experiments.** We have added the SSV2 evaluation results in Table 5 with the pre-trained model from K-400 datasets [C915]. The ablation results on K400 are shown in Table 12 [WGm9].

- **Model result analysis.** We have analyzed the strengths and weaknesses of our models and show some visualized examples in Figure 8, Figure 9, and Figure 10 of supplementary files.[bvLA].

---

### Meta-Review · Area_Chair_jjL9 · 2022-08-30

**Recommendation:** Accept
**Confidence:** Certain

**Metareview:**

This paper studies application of masked autoencoders to video data. It is a very empirical paper with lots of ablations and experiments. All three reviewers lean toward the acceptance of the paper. Reviewer C915 has a slight concern regarding the novelty of the paper over concurrent works including [50,53]. The reviewers believe that the ablation study is exhaustive and the paper has a good reproducibility. The authors are encouraged to add new experiments with kinetics pretraining in the final version.

**Award:**

No

---

### Decision · Program_Chairs · 2022-09-14

Accept